# Effects of Isoflavonoid and Vitamin D Synergism on Bone Mineral Density—A Systematic and Critical Review

**DOI:** 10.3390/nu15245014

**Published:** 2023-12-05

**Authors:** Miłosz Miedziaszczyk, Adam Maciejewski, Ilona Idasiak-Piechocka, Marek Karczewski, Katarzyna Lacka

**Affiliations:** 1Department of General and Transplant Surgery, Poznan University of Medical Sciences, 60-355 Poznan, Poland; m.miedziaszczyk@wp.pl (M.M.); ilonaidasiak@poczta.onet.pl (I.I.-P.); mkar@ump.edu.pl (M.K.); 2Department of Endocrinology, Metabolism and Internal Medicine, Poznan University of Medical Sciences, 60-355 Poznan, Poland; amaciejewski3@gmail.com

**Keywords:** phytoestrogens, isoflavonoids, vitamin D, bone mineral density

## Abstract

Phytoestrogens are non-steroidal plant compounds, which bind to α and β estrogen receptors, thereby causing specific effects. The best-known group of phytoestrogens are flavonoids, including isoflavonoids—genistein and daidzein. They play a role in the metabolism of bone tissue, improving its density and preventing bone loss, which contributes to reducing the risk of fractures. Vitamin D is found in the form of cholecalciferol (vitamin D3) and ergocalciferol (vitamin D2) and is traditionally recognized as a regulator of bone metabolism. The aim of this review was to evaluate the synergistic effect of isoflavonoids and vitamin D on bone mineral density (BMD). The MEDLINE (PubMed), Scopus and Cochrane databases were searched independently by two authors. The search strategy included controlled vocabulary and keywords. Reference publications did not provide consistent data regarding the synergistic effect of isoflavonoids on BMD. Some studies demonstrated a positive synergistic effect of these compounds, whereas in others, the authors did not observe any significant differences. Therefore, further research on the synergism of isoflavonoids and vitamin D may contribute to a significant progress in the prevention and treatment of osteoporosis.

## 1. Introduction

Estrogens are well-known players in bone metabolism in both females and males [1,2]. Moreover, estrogen deficiency is among the major factors leading to BMD decrease in postmenopausal women. These substances interact with osteoclasts and osteoblasts through multiple direct and indirect pathways, leading to an increase in osteoblast to osteoclast ratio [2]. However, estrogen or estrogen–progesterone therapies in postmenopausal women may be associated with potentially serious adverse reactions (cardiovascular events, stroke, venous thromboembolism, increase in breast cancer risk), and prescribing hormone replacement therapy solely to prevent osteoporosis remains controversial [3]. Phytoestrogens are plant-based non-steroidal substances that share structural similarities with 17-β-estradiol. Therefore, they have the ability to bind to estrogen receptors α (ERα) and β (ERβ), hence can act as agonists or antagonists of these receptors, competing with estradiol for the binding site. Phytoestrogens can exert their effects as antagonists of estrogen receptors with persistently high concentrations of estrogens (estradiol) in the body, which characterizes the period before menopause. In the case of low estrogen (estradiol) concentrations that occur in postmenopausal women, they can act as agonists [4,5,6]. Phytoestrogens represent the non-essential nutrients, since their absence in the diet does not result in disease. The best-known group of phytoestrogens are flavonoids, which include flavonols, quercetin, or isoflavonoids, genistein and daidzein. Phytoestrogens are found in plants from various families, including grasses (Gramineae), legumes (Papilionaceae), crucifers (Cruciferae), nightshades (Solanaceae) and cucurbits (Cucurbitaceae). Thus, estrogenic activity has been detected in a wide range of foods, including wheat, rice, soybeans, cabbage, apples, carrots, garlic, potatoes and coffee [7]. It has been observed that the content of phytoestrogens varied depending on the species, part of the plant, geographical and weather conditions prevailing in a given period [8]. Soy is the main source of isoflavones in the diet. Isoflavonoids can occur in an inactive form, bound as glycosides: genistin and daidzin. Present in food or added in the course of processing, the glycosides are hydrolysed and active aglycones (genistein and daidzein) are formed under the influence of gastric hydrochloric acid and bacterial intestinal microflora non-specific β-glucosidases [9]. Isoflavonoids play a role in the metabolism of bone tissue, improving its density, preventing bone loss, which contributes to reducing the frequency of fractures. It has been established that soy isoflavones have the capacity to both enhance bone growth and reduce bone resorption. The effect of isoflavonoids on bone mineral density (BMD) depends on their daily intake. Based on a systematic review, effective doses vary and a cut-off cannot be unequivocally identified, although efficacy has been observed in the range of 30–126 mg of soy isoflavones consumed per day. Another vital aspect is the duration of therapy. The period necessary for the occurrence of significant changes has been estimated at 6–18 months. However, compared to a 2-year research with a lower dosage, a 6-month study with a greater dose could demonstrate a stronger protective effect [10].

Vitamin D can be found as cholecalciferol (vitamin D3) and ergocalciferol (vitamin D2). Vitamin D3 can be found in animal food and is also produced in the skin when exposed to ultraviolet B (UV-B) radiation from the precursor 7-dehydrocholesterol. In turn, vitamin D2 is present in UV-irradiated mushrooms and yeast (subject to prior exposure to ultraviolet light) [11]. Vitamin D is traditionally recognized as a regulator of calcium and phosphorus homeostasis and bone metabolism [12].

In ovariectomized rats, isoflavonoids combined with vitamin D3 synergistically increased the expression of osterix transcription factor, and in bone cell cultures, when added to 1α,25(OH)2D3, i.e., the active form of the vitamin, they showed additive effects on the proliferation of preosteoblasts [13]. Furthermore, in the Japanese quail model, a combined supplementation with 25-OH-D3 and soy isoflavones synergistically improved BMD [14]. Based on the research results mentioned above, it can be hypothesized that the positive effect of both isoflavonoids and vitamin D may have a synergistic effect on bone metabolism, BMD and fracture risk, thus enhancing the effectiveness of their preventive and even therapeutic effects [13,14,15]. Bearing in mind the aforementioned findings, the purpose of this review was to evaluate the synergistic effect of isoflavonoids and vitamin D on BMD.

## 2. Materials and Methods

The MEDLINE (PubMed) Scopus and Cochrane databases were searched independently by two authors. The main search concept involved a combination of the terms (keywords) ’bone mineral density’ OR ’bone’ AND ’vitamin D’ AND ’phytoestrogens’ OR ’isoflavonoids’ OR ’isoflavones’ OR ’genistein’ OR ’daidzein’ OR ’cumestrol’ OR ’formononetin’ OR ’glycitein’ OR ’biochanin a’. The inclusion criterion was the information contained in the original and review papers, which referred to the data selected by the authors (original research articles as well as reviews articles obtained through the search strategy were included). In recent years, only few studies on the topic have been published; hence, it was decided not to apply any time criterion when searching the literature for our review. The protocol for the study has been registered as ID CRD42023480278. The PRISMA flowchart is presented in Figure 1.

## 3. Isoflavonoids

Isoflavones show their pharmacodynamic mechanism through preferential binding to the ERβ. Simultaneously, their nature of action as agonists or antagonists of these receptors is crucial. Depending on the tissue, ER expression may vary; therefore, phytoestrogens in bone tissue mimic the action of estrogens, and in breasts, they block the ERs, which may prevent the side effects of estrogen therapy (breast pain, breast tenderness) [16]. The effect of isoflavones on bones is based on a direct effect on bone cells. A meta-analysis is needed to determine whether isoflavones affect calcium absorption, as current reports on this topic are conflicting [17,18,19,20]. The phytoestrogens induce osteoprotegerin (BGLAP) production by binding to ERβ in osteoblastic cells in vitro. BGLAP, in turn, competes with the receptor activator for nuclear factor κB ligand (RANKL) and prevents preosteoclast maturation and thus resorption. Daidzein and genistein decrease osteoclastogenesis and consequently bone resorption via promoting transcription and ERβ binding. As a result, soy phytoestrogens improve bone health in postmenopausal women [21]. Figure 2 shows the classification of phytoestrogens.

### 3.1. Impact of Isoflavonoids on Bone Mineral Density

Original studies evaluating the efficacy of isoflavonoids on bone health are summarized in Table 1. Marini et al. evaluated the effect of genistein aglycone on bone following 3 years of therapy in a randomized controlled trial (RCT) (n = 138 women with postmenopausal osteopenia). Individuals were given 54 mg of genistein aglycone (n = 71) or a placebo (n = 67) per day. Calcium and cholecalciferol were administered in therapeutic levels to both treatment groups. When compared to the placebo group, genistein caused a greater increase in BMD in the lumbar spine and femoral neck at 36 months. The levels of bone-specific alkaline phosphatase (bALP), insulin-like growth factor I (IGF-I) and BGLAP were increased by genistein, whereas pyridinoline, a serum carboxy-terminal cross-linking telopeptide, and soluble nuclear factor kappa-light-chain-enhancer of activated B cells (NF-κB) were significantly reduced [22].

In an RCT, Tai et al. assessed the 2-year impact of soy isoflavones (172.5 mg genistein + 127.5 mg daidzein) on bone metabolism in postmenopausal women (n = 431) between the ages of 45 and 65. Dual-energy X-ray absorptiometry was used to evaluate the BMD of the proximal femur and lumbar spine at baseline and semi-annually thereafter. The difference in the mean percentage changes in BMD over the course of the treatment period was not statistically significant between the groups, even though relatively high doses of soy isoflavones were used and the intervention group’s serum concentrations of isoflavone metabolites, genistein and daidzein were significantly elevated after receiving isoflavones [23]. Another study also investigated the relationship between daidzein intake and BMD. Elderly patients (n = 100) with osteoporosis of the femur received a standard dietary supplement containing, i.a., 90 mg of isoflavones, or placebo for 12 months. No differences were observed between the groups in terms of the changes in BMD or between daidzein and the changes in bone mineralization [24].

An earlier RCT on the effects of genistein in postmenopausal women with femoral neck density < 0.795 g/cm^2^ was post hoc analyzed by Arcoraci et al. Women were divided into two groups: the study group (n = 62) received 54 mg/day of genistein in addition to 1000 mg of calcium and 800 IU of vitamin D3 as a placebo. The study group had 31.3% of the participants with first osteoporosis, compared to 30.9% in the control group, based on the femoral neck T-score scale. According to the 10-year fracture risk assessment (FRAX), the study group had a higher risk of hip fracture (4.2 ± 2.1) than the control group (4.1 ± 1.9). Femoral neck BMD increased from 0.62 g/cm^2^ at baseline to 0.70 g/cm^2^ after 2 years in the treatment group and decreased from 0.61 g/cm^2^ at baseline to 0.57 g/cm^2^ after 2 years in the control group. After 24 months, only 18 women in the study group suffered from osteoporosis, while in the control group, the number of postmenopausal women with osteoporosis did not change [25]. This study demonstrated a beneficial effect of adding phytoestrogen to standard vitamin D + calcium treatment in postmenopausal osteoporosis.

Cui et al. analyzed the relationship between osteoporotic fractures and dietary soy isoflavones (n = 48,584). The study involved post-menopausal women aged 43–76. An osteoporotic fracture occurred in 4.3% of the patients after a median follow-up of 10.1 years. A high soy isoflavone intake was linked to a lower incidence of osteoporotic fractures in women who had previously broken bones; HR = 0.72 (95% CI 0.55–0.93) was found for the highest (>42.0 mg/d) vs. lowest (<18.7 mg/d) intake quartile [26].

In the available scientific sources, there are five meta-analyses addressing the effect of isoflavones on BMD [27,28,29,30,31]. The authors of the first meta-analysis assessed urinary deoxypyridinoline (DPyr), which was selected as a marker of bone resorption, and bALP, a marker of bone formation. Nine studies were reviewed (n = 432). Urinary DPyr decreased by 2.08 nmol/mmol (95% CI: −3.82 to −0.34 nmol/mmol), while serum bALP increased by 1.48 µg/L (95% CI: 0.22–2.75 µg/L) in individuals ingesting isoflavones compared to those who did not consume them. The meta-analysis included studies in which various doses of consumed isoflavones were used (37.3 mg–118 mg); the studies lasted between 4 to 48 weeks [27]. Another meta-analysis analyzed 10 DPyr studies (n = 887), 10 bALP studies (n = 1210) and 8 BGLAP studies (n = 380). Patients received an average of 56 mg of soy isoflavones (aglycone equivalent) per day for 10 weeks to 12 months. DPyr concentration decreased in the study group as compared to the placebo group by −18.0% (*p* = 0.0007). Moreover, daily supplementation with approximately 80 mg of soy isoflavones for up to 12 months slightly increased bALP by 8.0% (*p* = 0.20) and BGLAP by 10.3% (*p* = 0.13) compared to the placebo group [28]. In another meta-analysis, it was demonstrated that the consumption of soy isoflavones resulted in significant improvements in BMD of the lumbar spine (0.76%; *p* = 0.03), hips (0.22%; *p* = 0.04) and the femoral neck (2.27%; *p* < 0.001). Subgroup analysis indicated that the improvement was significant in those with a normal body weight and in case of interventions longer than a year. Among the markers of bone turnover, substantial changes in the concentrations of pyridinoline (Pyr) and *C*-telopeptide were noted, although the results for the concentrations of BGLAP and bALP were statistically insignificant. Subgroup analysis of bone markers showed that isoflavones were more effective in individuals suffering from overweight/obesity [29]. Subsequent meta-analyses provide similar results and conclusions [30,31]. In view of these observations, it can be generally argued that soy isoflavone consumption reduces bone loss associated with menopause, although the effect is modified by different factors, including the dose used and duration of this diet.

The available scientific sources provide a study evaluating the synergistic effect of isoflavones and progesterone. Postmenopausal women with osteoporosis or presenting at least three risk factors for osteoporosis were randomized to group I (n = 23) receiving soy milk containing isoflavones—76 mg, group II (n = 22) where transdermal progesterone was administered, group III (n= 22) in which the participants received both isoflavone-rich soy milk and progesterone and group IV (n = 22), which was the placebo group (consuming isoflavone-poor soy milk, progesterone-free cream). BMD values were measured in the lumbar spine and hip using X-ray absorptiometry at double energy (DEXA) at baseline and then after 2 years. The percentage change in the lumbar spine BMD was (+1.1%)—group I, (+1.1%)—group II, (−4.2%)—group III and (−2.8%)—group IV. There were no significant changes in the BMD of the femoral neck. Therefore, it is evident that phytoestrogens or transdermal progesterone positively affected BMD, although the combination of these therapies did not induce beneficial effects [32].

Another meta-analysis assessed the impact of isoflavones on the mineral density of bone mass. Based on 63 RCTs (n = 4754) and a placebo group (n = 4272), isoflavone interventions significantly improved lumbar spine BMD (MD = 0.0175 g/cm^2^; 95% CI, 0.0088 to 0.0263, *p* < 0.0001) and femoral neck (MD = 0.0172 g/cm^2^; 95% CI, 0.0046 to 0.0298, *p* = 0.0073) in postmenopausal women. A subgroup analysis revealed that an isoflavone intervention improved BMD when the therapy lasted for at least 12 months and contained genistein at a dosage of at least 50 mg per day [33].
nutrients-15-05014-t001_Table 1Table 1Original studies assessing the efficacy of isoflavonoids supplementation on BMD/fracture risk. BMD—bone mineral density [22,23,24,25,26,34,35,36,37,38,39,40,41,42,43,44,45,46].AuthorYearCountryStudied GroupPatients/ControlsInterventionTimeBMD/Fracture Risk(+) Reduction Bone Loss/Reduction Fracture Risk(−) No Reduction Bone Loss/No Reduction Fracture RiskOther ParametersType of the StudyAgnusdei et al. [39]1997ItalyPostmenopausal women (65–79 y)41/43200 mg ipriflavoneControl group: 1 g calcium24 monthsForearm BMD (+)Not studiedRandomized, double-blind, placebo-controlledKhalil et al. [34]2002USAHealthy men (59.2 ± 17.6 y)24/2288 mg/day of isoflavones3 monthsNot studiedBone turnover markers, IGF-1Randomized, double-blind, controlledChen et al. [40]2003ChinaPostmenopausal women (48–62 y)68/671 g soy extracts and 80 mg isoflavones12 monthsFemoral neck BMD (−)lumbar spine BMD (−)Not studiedRandomized, double-blind, placebo-controlledBunout et al. [24]2006ChileElderly subjects (both genders) with femoral osteoporosis45/3990 mg/day of isoflavones (together with other supplements)12 monthsFemoral neck BMD (−)lumbar spine BMD (−)Bone turnover markers, 25(OH)D, PTH, osteocalcinRandomized, controlledNewton et al. [35]2006USAMen and women (50–80 y)9F, 45M/7F, 54M45.6 mg genistein, 31.7 mg daidzein, 5.5 mg glycitein daily12 monthsProximal femur BMD (−)lumbar spine BMD (+, only women)Not studiedRandomized, double-blind, placebo-controlledMarini et al. [38]2007ItalyPostmenopausal women with low femoral neck BMD (osteopenia or osteoporosis)198/19154 mg of genistein aglycone daily24 monthsFemoral neck BMD (+)lumbar spine BMD (+)Bone turnover markersRandomized, double-blind, placebo-controlledMarini et al. [22]2008ItalyWoman with postmenopausal osteopenia71/6754 mg of genistein aglycone daily36 monthsFemoral neck BMD (+)lumbar spine BMD (+)Bone turnover markers, sRANKL, osteoprotegrin, IGF-1Randomized, double-blind, placebo-controlledMarini et al. [41]2008ItalyWoman with postmenopausal osteopenia(49–67 y)198/19154 mg of genistein aglycone daily24 monthsNot studiedBone turnover markersRandomized, double-blind, placebo-controlledBrink et al. [42]2008NetherlandsPostmenopausal women (53 ± 3 y)119/118110 mg of soy isoflavones12 monthsLumbar spine BMD (−)Bone turnover markersRandomized, double-blind, placebo-controlledWong et al. [43]2009USAPostmenopausal women (40–60 y)134/134120 mg of soy hypocotyl aglycone isoflavonesControl group: without treatment24 monthsWhole body BMD (+)Bone turnover markersRandomized, double-blind, placebo-controlledAlekel et al. [44]2009USAPostmenopausal women (45.8–65.0 y)73/74120 mg of soy isoflavonesControl group: without treatment36 monthsFemoral neck BMD (−)lumbar spine BMD (−)Not studiedRandomized, double-blind, placebo-controlledTai et al. [23]2012Taiwan-ChinaPostmenopausal women (45–65 y)200/199172.5 mg genistein + 127.5 mg daidzein daily24 monthsProximal femur BMD (−)lumbar spine BMD (−)Bone turnover markersRandomized, double-blind, placebo-controlledChilibeck et al. [45]2013CanadaPostmenopausal women 76/73165 mg of soy isoflavones24 monthsLumbar spine BMD (−)Not studiedRandomized, double-blind, placebo-controlledArcoraci et al. [25]2017ItalyPostmenopausal women with femoral neck osteoporosis62/5954 mg of genistein aglycone daily24 monthsFemoral neck BMD (+)Bone turnover markersPost hoc analysis, randomized, double-blind, placebo-controlledZhang et al. [46]2020ChinaPostmenopausal women (40–55 y)38/3715 mg of soy isoflavones6 monthsBMD (−)Not studiedRandomized, double-blind, placebo-controlledCui et al. [36]2021ChinaMen (40–74 y)61,025population-based cohortSoy isoflavone intakeMedian follow-up ~9.5 yearsOsteoporotic fracture risk (−)Not studiedPopulation based, prospective, observationalCui et al. [26]2022ChinaPostmenopausal women (43–76 y)48,584 population-based cohortSoy isoflavone intakeMedian follow-up ~10 yearsOsteoporotic fracture risk (+, only in bone fracture history group)Non-osteoporotic fracture risk (−)Not studiedPopulation based, prospective, observational

### 3.2. Effects of Isoflavones in Men

Somjen et al. analyzed the response of human bone cells to phytoestrogens in pre- and post-menopausal women and men. The bone cells of premenopausal women showed a greater creatine kinase stimulation than those in postmenopausal women. In contrast, male cells did not respond to phytoestrogens [47]. Another study also investigated the effects of phytoestrogens on bone health in men. Healthy men (n = 64; 59.2 ± 17.6 years) were enrolled to receive 40 g of soy protein (test group) or 40 g of milk protein (control group) daily for 3 months. Men in the study group had greater levels of IGF-I (*p* < 0.01) than in the control group, which is linked to a higher rate of bone growth. Nevertheless, the authors do not provide the results in the form of numerical values. There were no significant differences seen between the groups in terms of serum alkaline phosphatase activity, bALP activity, indicators of bone production or urine excretion of deoxypyridinoline. Due to a significant decrease in bone density, which occurs in men around the age of 65, the data were analyzed separately for men under 65. There were also no significant differences between the groups [34].

In another study, 145 participants aged 50–80 years were subdivided into group I receiving a soy drink (83 mg isoflavones; 45.6 mg genistein, 31.7 mg daidzein) and group II (3 mg isoflavones). The study was conducted within a period of 12 months. BMD was assessed using dual-energy X-ray absorptiometry at the hip and anterior posterior spine (L1-L4). The mean percentage change in BMD in women was 0.58 ± 0.70% for group I vs. 1.84 ± 0.86% for group II (*p* = 0.05) in the spine examination; no significant change was found for men. In the case of the results in the hip bone examination, no significant difference was observed in women and men, and the results did not differ significantly between these groups [35].

Cui et al. investigated the association between soy isoflavone consumption with the risk of osteoporotic fractures in the Shanghai Men’s Health Study (n = 61,025). The study included men aged 40 to 74, who were surveyed for 9.5 years. In total, 1.2% and 3.4% of the participants, respectively, experienced osteoporotic or non-osteoporotic fractures. The authors reported that a high consumption of soy isoflavones (>45.2 vs. <21.7 mg/d) was associated with a 25% reduction in the risk of osteoporotic fractures (HR = 0.73, 95% CI 0.56–0.97) [36].

## 4. Vitamin D

Vitamin D and parathormone (PTH) are among the key factors regulating calcium–phosphate hemostasis and bone metabolism. The association between vitamin D and the skeletal system has been known for decades, although some aspects of this puzzle require further research. Vitamin D active metabolites (predominantly 1α,25(OH)2D) directly influence intestinal calcium and phosphate absorption, renal calcium and phosphate excretion as well as bone turnover/remodeling [48]. The kidneys are responsible for the hydroxylation of 25(OH)D to the active form of vitamin D (1α,25(OH)2D). 1.25(OH)2D is responsible for the proper intestinal absorption of calcium (TRPV6, calbindin D9k and PMCA1b). The correct concentration of 1α,25(OH)2D inhibits the synthesis of parathyroid hormone (PTH) by the parathyroid glands, regulating bone resorption processes. In addition, the active form of vitamin D is responsible for the reabsorption of calcium in the kidneys (TRPV5 and TRPV6, calbindin D28k and PMCA). The kidneys therefore play a key role in the process of calcium metabolism in the body. In the case of renal dysfunction, the active form of vitamin D is not synthesized. As a result, the concentration of calcium in the blood decreases due to impaired absorption of calcium in the intestines and an increase in the concentration of phosphates that bind calcium. A decrease in the concentration of calcium in the blood activates the parathyroid glands to produce parathyroid hormone; additionally, this hormone is synthesized as a result of the lack of an inhibition mechanism by 1α,25(OH)2D. PTH alone induces the hydroxylation of 25(OH)D to 1α,25(OH)2D, bone resorption and renal Ca reabsorption. Increased serum Ca levels provide negative feedback on PTH release, keeping serum Ca levels within a narrow time window, whereas excess Ca from absorption is deposited in the bones or excreted in the urine. Through this process, vitamin D can stimulate Ca absorption, which is a fundamental step in increasing BMD. Clinical studies have shown that the threshold level of 25(OH)D is 20 nmol/L to maximize vitamin D-mediated Ca absorption in adults [49,50]. See also Figure 3. These actions are modulated by multiple other factors (systemic, including PTH, calcium and phosphate level, estrogens, thyroid hormones; local growth factors: IGF-1, transforming growth factor β1 (TGF-β), fibroblast growth factor-23 (FGF-23), etc.; and cytokines) [51,52]. Although traditionally classified as a vitamin, vitamin D is mainly produced in the skin from 7-dehydrocholesterol as a result of sunlight (UVB radiation) exposure. Therefore, identifying vitamin D as a steroid hormone or prohormone would be more precise. Diet is a minor source of vitamin D and usually it is incapable of compensating low skin synthesis [53]. The rich dietary sources of vitamin D involve only certain fish types, mushrooms and fish liver oils; whereas a lower vitamin D content can be found in cheese, beef liver, eggs or chocolate [54]. In some countries, vitamin D food fortification programs are implemented with an observed improvement in 25(OH)D concentrations at a population level [55]. In order to achieve the most biologically active vitamin D metabolite—calcitriol (1α,25(OH)2D)—two enzymatic hydroxylation reactions are required: 25-hydroxylation (in the liver) and hydroxylation at position 1α (mainly in the kidneys). Serum 25(OH)D is usually used as an indicator of vitamin D status due to its long half-life. Vitamin D deficiency seems to be a common problem worldwide, both in sunlight-rich and deficient regions, which may stem from a variety of factors, including latitude, skin pigmentation, popularization of sunscreen use, lifestyle, dietary intake, etc. Additionally, the deficiency risk increases more with age or weight [56].

A lack of vitamin D can cause rickets in children and osteomalacia in adults. Furthermore, low vitamin D level is also associated with higher bone turnover, skeletal mineralization defects and, as a consequence, low bone mass, increased prevalence of osteoporosis and a high fracture risk [57,58]. In recent years, more evidence has emerged regarding the pleiotropic effects of vitamin D, including mainly immunomodulating and antiproliferative/antioncogenic effects. Therefore, vitamin D deficiency may also be associated with a higher prevalence of infectious, inflammatory (including autoimmune) and neoplastic diseases [59,60].

### 4.1. Impact of Vitamin D on Bone Mineral Density

Despite different recommendations from both national and international societies and the popularization of vitamin D supplements, vitamin D insufficiency or deficiency is still common in Europe (40.4% and 13.0%, respectively) [61,62,63]. In Poland, as many as 90% of adults may present vitamin D levels lower than 30 ng/mL (values may vary depending on the season) [64]. In the case of vitamin D deficiency, supplementation seems beneficial in a variety of aspects, including multiple extraskeletal effects, leading to a reduced mortality [65,66]. In terms of bone diseases, it protects from rickets/osteomalacia, suppresses excessive PTH secretion and regulates calcemia. Additionally, a positive correlation between vitamin D and BMD has also been reported [67,68]. However, as shown in the meta-analyses, benefits from further vitamin D supplementation are questionable when the baseline vitamin D level is sufficient [69]. Moreover, the U-shaped effect of vitamin D on bone health has also been suggested, whereby high doses may potentially produce adverse effects [70,71,72]. The reported inconsistencies may partly stem from differences between studies, including ethnic differences. In Asians, both results in favor and against the positive impact of vitamin D on BMD have been noted [57]. Interestingly, African Americans tend to have a higher BMD, despite lower vitamin D concentrations in comparison with white Americans. In fact, vitamin D supplementation among this group failed to show any evident benefits [73]. Some authors suggest that sufficient vitamin D level may be necessary, but not enough to prevent BMD loss [74]. In addition to all the above-mentioned issues, the primary question and end-point of most studies was the potential of vitamin D to protect against fractures. Various analyses demonstrated that vitamin D supplementation exerted a protective effect on fracture risk [75,76], although also opposite results were reported [77,78,79]. In a recent systematic umbrella review, the authors concluded that vitamin D alone did not significantly reduce fracture risk. However, vitamin D together with calcium could be beneficial (hip fracture reduction and, to a lesser extent, any fracture reduction) [80]. Nevertheless, it is also essential to account for the underlying causes of the variability between the studies, which groups of patients benefit as well as for the exact dose of supplementation. These partially disappointing results with regard to how vitamin D affects BMD and fracture risk enhance the need to search for other factors supporting BMD maintenance.

Notably, not only the level of vitamin D, but also vitamin D-related gene variants may contribute to BMD and bone diseases. These genes include VDR (vitamin D receptor), CYP2R1 (25-hydroxylase), CYP24A1 (24-hydroxylase), CYP27B1 (1α-hydroxylase), GC/DBP (vitamin D binding protein) and DHCR7 (7-dehydrocholesterol reductase) [81]. Predominantly VDR gene SNPs (single nucleotide polymorphisms) have been studied to date. A recent meta-analysis of fourteen studies reported that BsmI and FokI VDR polymorphisms were associated with lower BMD in men [82]. VDR gene BsmI polymorphism–BMD correlation was also confirmed in pediatric patients [83]. In the female group, different reports also pointed to the potential role of BsmI VDR SNP in BMD regulation [84,85]. Different VDR polymorphisms (BsmI, ApaI, TaqI, FokI) were shown to impact the risk of osteoporosis, with ethnic differences also observed [86,87]. Nonetheless, between-studies inconsistency was also present in this aspect [88].

### 4.2. The Synergistic Effects of Isoflavonoids and Vitamin D

Original studies evaluating the synergistic efficacy of isoflavonoids and vitamin D on bone health are summarized in Table 2. In the available scientific literature, there are only five studies assessing the effect of the combined effect of isoflavonoids and vitamin D on BMD. Ushiroyama et al. found a significant reduction in vertebral bone loss after 18 months in patients receiving combined therapy by 3.70% compared to the control group, *p* < 0.001. Lappe et al. conducted a 6-month study (n = 70). In the study group (n = 30), there were no changes in the BMD values of the femoral neck, whereas in the placebo group (n = 28), the BMD values decreased considerably. Moreover, bALP and *N*-telopeptide were significantly elevated in the study group compared to the placebo group [89]. The limitations of the study are short duration, relatively small groups and multiple dietary factors used in the study group what preclude the possibility of the evaluation of the role of each of them. Based on these two studies, it can be concluded that the combined intake of isoflavonoids with vitamin D inhibits bone loss in postmenopausal women.

An animal-model research was carried out in order to assess the synergistic impact of vitamin D and isoflavonoids. Ovariectomized female rats (n = 50) were subdivided into three groups, i.e., the control group, a group receiving only vitamin D (2400 IU/kg) and the group where vitamin D, resveratrol, quercetin and genistein were administered for 8 weeks. Treatment (vitamin D + 400 mg/kg resveratrol + 2000 mg/kg quercetin + 1040 mg/kg genistein) significantly reduced weight gain. This therapy significantly prevented trabecular bone loss and reduced the number of adipocytes and osteoclasts in the bone marrow compared to the control group and the group receiving only vitamin D. Therefore, the synergistic effect of the combination of genistein and vitamin D may effectively reduce bone loss and weight gain in postmenopausal women [90]. In another study, also involving the animal model, the authors showed that calcium and cholecalciferol supplementation did not increase the effect of genistein on improving bone loss indicators [91].

It has been demonstrated that genistein increases transcription of the VDR gene promoter [92,93]. Furthermore, concomitant treatment with the antiestrogen tamoxifen (TAM) showed that the effect of phytoestrogens on the VDR promoter is estrogen receptor dependent [93]. It is of note that genistein increased the expression of vitamin D hydroxylase CYP27B1 and inhibited the expression of CYP24 in colon cancer cells [47], although it inhibited the expression of vitamin D hydroxylase CYP24 and CYP27B1 in prostate cells [94]. In another study, calcitriol was shown to induce the expression of estrogen receptor α through the direct transcription regulation and epigenetic modifications in estrogen receptor-negative breast cancer cells [95].

The proposed mechanisms of synergistic action of isoflavonoids and vitamin D are presented in Figure 4.

In recent years, the influence of the immune system on bone metabolism has been emphasized, including immune cells’ influence on RANKL and the role of Th17 lymphocytes, IL-17 and TNF in osteoclastogenesis [97,98]. In fact, vitamin D is a well-known immunomodulating factor [88]. Additionally, phytoestrogens acting through estrogen receptors may also modulate the microenvironment of bone cells by mimicking the known estrogen action on immune system [2]. The aforementioned aspect should constitute the area of further research investigating another potential mechanism of vitamin D., i.e., phytoestrogen interaction.
nutrients-15-05014-t002_Table 2Table 2Original studies assessing the efficacy of isoflavonoids + vitamin D supplementation on BMD/fracture risk. BMD—bone mineral density [89,99,100,101,102].AuthorYearCountryStudied GroupPatients/ControlsInterventionTimeBMD/Fracture Risk(+) Reduction Bone Loss/Reduction Fracture Risk(−) No Reduction Bone Loss/No Reduction Fracture RiskOther ParametersType of the StudyUshiroyama et al. [100]1994JapanPostmenopausal women (45–65 y)20/35Ipriflavone 600 mg/day, alfacalcidol 1 µg/day Control group: without treatment18 monthsVertebral BMD (+)Bone turnover markersRandomized, placebo-controlledLappe et al. [89]2013USAPostmenopausal women (45–55 y)30/28Genistein 30 mg/day and vitamin D3 800 IU/day (together with other supplements)Control group: vitamin D3 800 IU/day (together with other supplements)6 monthsFemoral neck BMD (+)Ward’s triangle BMD (+)lumbar spine BMD (−)Bone turnover markersRandomized, double-blind, placebo-controlledBevilacqua et al. [101]2013ItalyPostmenopausal women (36–84 y)28 for study group, without control group Soy isoflavones 40 mg/day, inulin 3 g/day, vitamin D 300 IU/day, calcium 500 mg/day3 monthsNot studiedBone turnover markersRetrospective studyPerez-Alonso et al. [102]2017SpainPostmenopausal women (55 ± 4 y)102 in 2 groups (no more detailed information)Genistein 90 mg/day and vitamin D3 800 IU/day and calcium 1000 mg/dayControl group: vitamin D3 800 IU/day and calcium 1000 mg/day3 monthsNot studiedBone turnover markersRandomized, double-blind, placebo-controlledPerez-Alonso et al. [103]2023SpainPostmenopausal women (55 ± 4 y)150 in 3 groups (no more detailed information)Genistein 90 mg/day and vitamin D3 800 IU/day and calcium 1000 mg/dayControl group: vitamin D3 800 IU/day and calcium 1000 mg/daySecond control group: without treatment3 monthsNot studiedBone turnover markersRandomized, double-blind, placebo-controlled

## 5. Conclusions

Phytoestrogens acting through the ER seem to be a potentially interesting nutrient or supplement in the prevention of osteoporosis, particularly in the population of postmenopausal women. Based on a meta-analysis conducted this year, it can be concluded that isoflavones, specifically those containing genistein in an amount of at least 50 mg/day, can effectively increase BMD in postmenopausal women.

Nevertheless, phytoestrogen-rich diet has also raised some concerns, as it has been identified as one of the endocrine disrupting factors. Initially, its safety for estrogen receptor-positive breast cancer patients was questioned. Yet, further studies largely resolved these concerns [103]. Moreover, in a group of premenopausal women, a soy-rich diet may potentially reduce the risk of breast cancer [104].

Simultaneously, reports of other benefits of these plant-based agents than just bone health have emerged, predominantly in postmenopausal women. These include cardiovascular risk reduction [105], insulin sensitivity increase along with reduction of fasting glucose [106], positive effect on homocysteine and HDL levels [107] and reduction of total cholesterol [108].

Age-related BMD loss, which begins after reaching peak bone mass, is a natural process that can be modulated by comorbidities, malnutrition, etc. Different dietary interventions are being currently tested (vitamin D, calcium, other minerals such as magnesium, potassium, protein intake, polyunsaturated fatty acids) [109]. Synergistic effect is expected, although RCTs are lacking. The scientific literature regarding vitamin D and phytoestrogen synergism is scarce with no recommendations available, despite certain physiological background to support the benefits of their combined use. Further large scale and well-designed studies are needed to establish clear recommendations.

Reference publications do not provide consistent data with regard to the synergistic effect of isoflavonoids on BMD. In fact, two studies demonstrated the positive synergistic effect of these compounds, whereas according to other authors, no significant differences were observed. However, the available scientific data emphasize the fact that isoflavonoids affect BMD, and the same conclusions apply to vitamin D. Further research on the synergism of isoflavonoids and vitamin D may contribute to significant advances in the prevention and treatment of osteoporosis. Finally, studies involving the premenopausal women population as well as the male population are lacking and represent other areas worth investigating.

## Figures and Tables

**Figure 1 nutrients-15-05014-f001:**
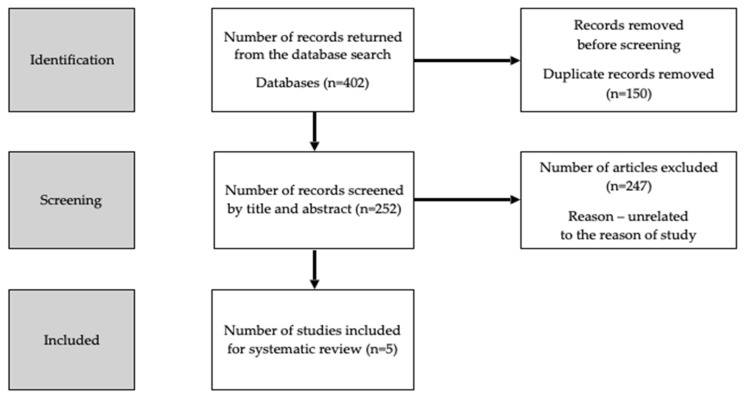
PRISMA Chart for the systematic review.

**Figure 2 nutrients-15-05014-f002:**
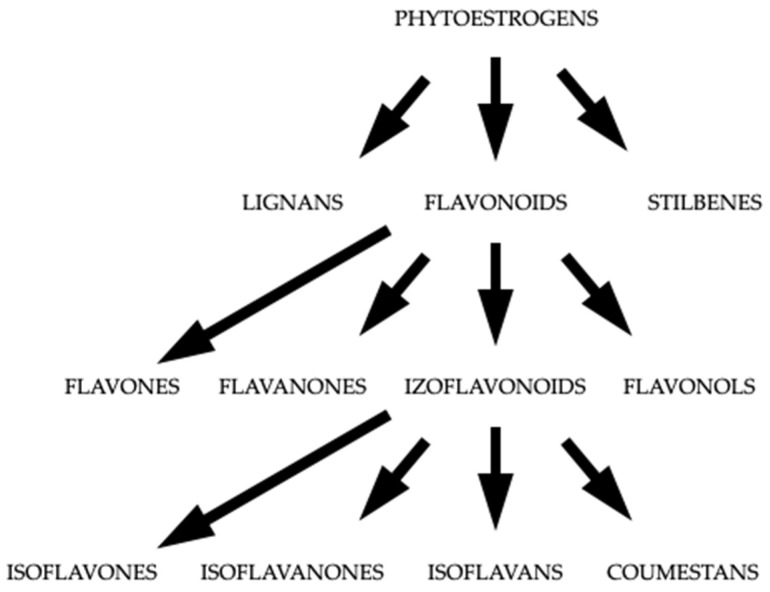
The classification of phytoestrogens.

**Figure 3 nutrients-15-05014-f003:**
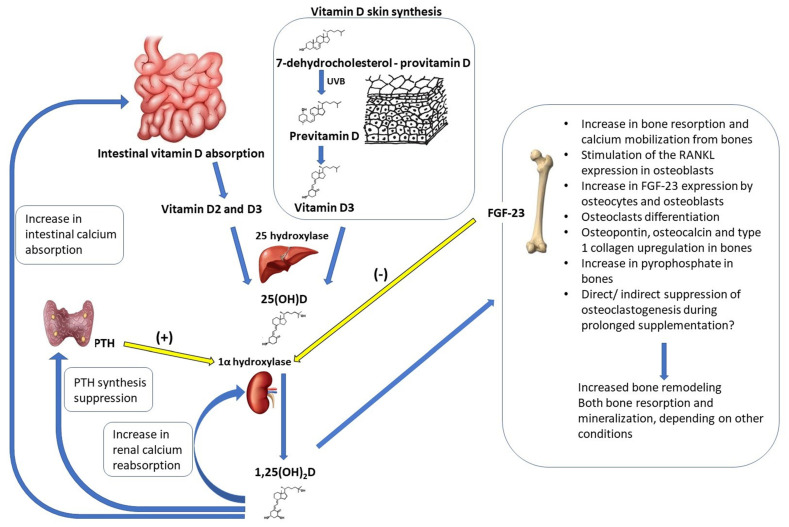
Impact of vitamin D on bone mineral density—a graphic representation.

**Figure 4 nutrients-15-05014-f004:**
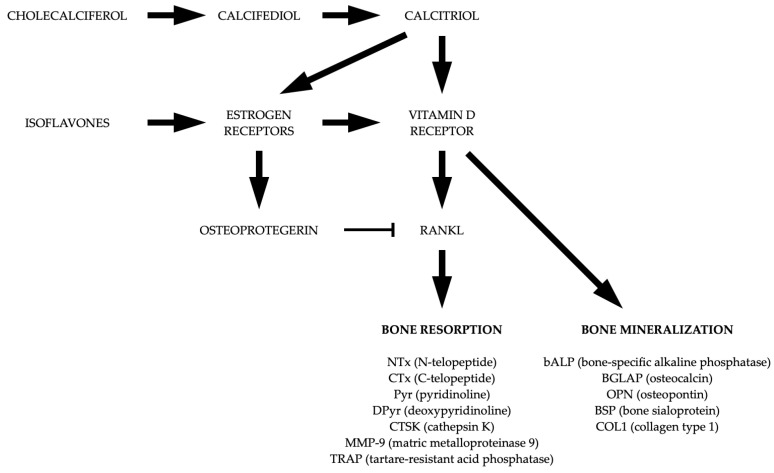
Proposed mechanisms of the synergistic effects of isoflavonoids and vitamin D [12,13,14,15,16,17,18,19,20,21,22,23,24,96].

## Data Availability

The data presented in this study are available on request from the corresponding author.

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
