# Peer review of "Effects of Isoflavonoid and Vitamin D Synergism on Bone Mineral Density—A Systematic and Critical Review"

_nutrients, 2023, doi:10.3390/nu15245014_

Round 1
Reviewer 1 Report (Previous Reviewer 2)
Comments and Suggestions for Authors
The authors have addressed the criticisms and suggestions made in my original review of this manuscript. There is still some wording which is misinforming at line 136, where mushrooms are listed as a dietary source of vitamin D. This should be modified to read "UV-irradiated mushrooms".
Comments on the Quality of English LanguageThe English expression is largely well done. There are a few places where sentence structure could be improved.
Author Response
Dear Reviewer,
thank you very much for the effort put into the review and for valuable comments. The comments helped us significantly improve the article. All changes are marked in red in the manuscript.
Yours Sincerely

Reviewer 2 Report (Previous Reviewer 1)
Comments and Suggestions for Authors
The authors have answered the comments appropriately
Author Response
Dear Reviewer,
thank you very much for the effort put into the review and for valuable comments. The comments helped us significantly improve the article.
Yours Sincerely
This manuscript is a resubmission of an earlier submission. The following is a list of the peer review reports and author responses from that submission.
Round 1
Reviewer 1 Report
Comments and Suggestions for Authors
Synergistic effects of phytoestrogens & vitamin D on bone mineral density
The authors generally abide to the proposed guidelines for a critical review. However, the text needs improvement in the syntax and would benefit from a revision in terms of structure and language. Information given being generally generic, the different topics require to be described more in depth. Examples are given below.
The abstract concisely identifies the research question and provides the justification for its need.
The introduction although providing valid information should be edited to be clearer and provide more balanced and in depth information.
Page 1: The authors mention that phytoestrogens have the ability to bind to estrogen receptors and that they act as agonist/antagonists. These statements need to be supported by references and authors should describe more specifically their mechanisms of action.
Page 2: The sentence “They can be present in a bound form as glycosides: genistin and daidzin, where the aglycone is genistein and daidzein, with the aglycones released by the intestinal microflora exhibiting estrogenic activity [4]” is such an example. Although the sentence infers that the aglycones have the estrogenic activity, the way written leads to think that it is the intestinal microflora that has this activity.
Page 2: Correct the typographical error “animal foo” to “animal food”
Page 2: The authors mention that isoflavonoids play a role in bone metabolism citing reference 5. This reference, a systematic review on the subject, provides ample information. The authors should develop this topic.
Page 2: What entitles the author to state that “it is expected the positive effects of both isoflavonoids and vitamin D on bone be synergistic? This a biased statement. Is it not the purpose of this critical review to evaluate whether these 2 classes of compounds act synergistically?
Page 2: The methods section also requires clarification. What do the authors mean by the general statement “The browsing (should be search) strategy included controlled vocabulary and keywords”?
What do the authors want to convey as a message in “The inclusion criterion was the information contained in the original and review articles, which referred to the data selected by the authors.”? Is it that original research articles as well as reviews articles obtained through the search strategy were included?
Page 2: The section on isoflavonoids needs revision. In the same sentence “Due to the effect on ERβ, there is an increased risk of breast cancer, however this therapy improves bone health in postmenopausal women [9] ” the authors mention effects of isoflavonoid on risks of breast cancer and improvement of bone health in postmenopausal women citing refence 9, a review that addresses the interaction of vitamin D & isoflavones, the subject of the present review. The also mention that these isoflavones act on bone cells & not calcium absorption. This and what follows need to be better described and substantiated with references.
Overall, this review needs to be revised to provide the information in a better fashion.
Author Response
Dear Reviewer,
The authors thank you very much for the effort put into the review and for valuable comments. The comments helped us significantly improve the article.
All changes are marked in red and blue in the manuscript.
Kind Regards
Authors

Reviewer 2 Report
Comments and Suggestions for Authors
Many published studies attempting to identify effects of vitamin D status on various biological and disease incidence measurements have provided either inconclusive or conflicting conclusions. This review of the literature attempting to discern whether there is evidence for a synergistic effect of good vitamin D status and intake of various isoflavonoids comes to a similar conclusion that evidence for a possible beneficial interaction between the oral intake of these plant substances and vitamin D status has not been established. Nevertheless, it is worthwhile for the authors to review the current published findings and to explain in a review the biological information that suggests an hypothesis of beneficial interaction on bone mineral density.
However, the way the published findings are reviewed in this manuscript makes it very difficult to relate what has been found in publications to the general hypothesis. For a start, it is confusing to have a shoulder heading of “Results”. This word is usually used for original research observations and not for a review of published work. The review would be more easily understood if the shoulder headings indicated each component of the wider hypothesis. A further difficulty is taking each publication and summarizing the findings. This makes the manuscript simply a catalogue of publications rather than an authoritative review expressing the authors’ expertise on the broad topic. It would make the manuscript more easily understood if the authors were to write about the different aspects of the hypothesis and simply quote papers that support or do not support the concepts explained in this overall review. Some reviews of different studies compare the findings in a table which then summarizes how different studies either agree or differ for each variable or measurement. As the manuscript stands, the narrative flow is not easy to follow.
Author Response

(The authors gave the same response as above.)

Round 2
Reviewer 1 Report
Comments and Suggestions for Authors
The authors have improved the original manuscript. There are still some modifications that are suggested below.
Page 2: The sentence “Under the influence of gastric hydrochloric acid and non-specific β-glucosidases of bacterial intestinal microflora, present in food or added during processing, the glycoside is hydrolysed and an active aglycone is formed, where the aglycones are genistein and daidzein.” should be slightly modified to “Present in food or added during processing, the glycosides are hydrolysed and active aglycones (genistein and daidzein) formed under the influence of gastric hydrochloric acid and bacterial intestinal microflora non-specific β-glucosidases.”
Page 2: The sentence “The synergistic effect of vitamin D and isoflavonoids increased the expression of the osterix transcription factor and the proliferation of preosteoblasts when combined with vitamin D3 [13].” Should be modified to “In ovariectomized rats, isoflavonoids combined vitamin D3 synergistically increased the expression of osterix transcription factor and, in bone cell cultures, added to 1a,25(OH)2D3, the active form of the vitamin, they showed additive effects on the proliferation of preosteoblasts [13].”
Page 2: The sentence “In another study, also in an animal model, bone mineral density was improved by the combined supplementation of 25-OH-D(3) and soy isoflavones [14].” Should be modified to “Furthermore, in the Japanese quail model, combined supplementation with 25-OH-D3 and soy isoflavones synergistically [improved bone mineral density 14].”
Page 10: The sentence “Moreover, in premenopausal women group soy-rich diet may potentially reduce the risk of breast cancer [91]” could be modified to “Moreover, in a group of premenopausal women a soy-rich diet may potentially reduce the risk of breast cancer [91]”
Author Response
Dear Reviewer,
The authors thank you very much for the effort put into the review and for valuable comments. The comments helped us significantly improve the article.
Kind Regards
Authors

Reviewer 2 Report
Comments and Suggestions for Authors
The authors have improved the readability of the manuscript. However, it is still difficult for a reader to understand the state of knowledge overall from the published findings. There are a series of paragraphs which simply catalogue the results presented in various papers. Each paragraph starts as follows:
Morini H. et al
Tai T.Y. et al
Arcoraci V. et al.
Cui et al
Somjen D. et al
Lappe J. et al
It would be much easier for a reader to understand the relative results of each of these studies if the findings were presented as a Table rather than as individual summaries. The important feature of literature reviews is the way a range of publications can be easily understood by a reader. In the current format a reader actually has to undertake the process of analysis from the catalogue of summaries of each publication.
On page 2 of the manuscript there are two errors of description:
1. “Vitamin D is a fat-soluble steroid hormone” This is not correct. Vitamin D is a steroid hormone precursor. It has to be metabolically converted to a hormone.
2. “In turn, vitamin D2 is contained in mushrooms and yeasts” This also is misleading. Vitamin D2 is only present in these fungal species after they are irradiated with ultraviolet light. If they are not exposed to ultraviolet light, they contain no vitamin D2.
Author Response

(The authors gave the same response as above.)
